# α-Glucosidase and Pancreatic Lipase Inhibitory Activities of Diterpenes from Indian Mango Ginger (*Curcuma amada* Roxb.) and Its Derivatives

**DOI:** 10.3390/molecules24224071

**Published:** 2019-11-10

**Authors:** Yuri Yoshioka, Naori Yoshimura, Shinichi Matsumura, Hiroto Wada, Maya Hoshino, Shouhei Makino, Masanori Morimoto

**Affiliations:** 1Natural products, Inabata Koryo Co., Ltd., Osaka 320027, Japan; y-yoshioka@inabatakoryo.co.jp (Y.Y.); shin-matsumura@inabatakoryo.co.jp (S.M.); makino-s@inabatakoryo.co.jp (S.M.); 2Department of Applied Biological Chemistry, School of Agriculture, Kindai University, Nara 6318505, Japan; 1933670018j@nara.kindai.ac.jp (N.Y.); bontarou1004@i.softbank.jp (H.W.); 1311430002s@nara.kindai.ac.jp (M.H.)

**Keywords:** *Curcuma amada* Roxb., diterpene, α-glucosidase inhibitor, pancreatic lipase inhibitor, anti-obesity

## Abstract

Enzymatic inhibitions of crude extracts and their constituents from Zingiberaceae against both rat intestinal α-glucosidase and porcine pancreatic lipase were investigated. Structure–activity relationships using their derivatives were also investigated. The rhizomes extract of mango ginger, *Curcuma amada* showed remarkable inhibitory activity in the screening test. Two natural labdane diterpenes **1** and **2** and a drimane sesquiterpene **3** were major constituents isolated from this hexane extract. Among them, (*E*)-labda-8(17),12-diene-15,16-dial (**1**) was the most prominent compound and showed inhibitory activity against both α-glucosidase and lipase. Derivatives **4**–**10** from compound **1** were prepared and evaluated using inhibitory assays with these enzymes. The reduced derivative **4** maintained α-glucosidase inhibitory activity, but had decreased pancreatic lipase inhibitory activity compared with parent compound **1**. Other tested derivatives of compound **1**, including acetates **5**–**7** and oxidative derivatives **8–10**, had very weak α-glucosidase inhibitory activity. Most of these compounds showed moderate pancreatic lipase inhibitory activity. However, only sesquiterpene albicanal (**3**) showed drastically decreased pancreatic lipase activity compared with **1**. These findings suggested that molecular size was essential for enzymatic inhibitory activities of these compounds. These results demonstrated that mango ginger may be useful for the prevention of obesity and being overweight.

## 1. Introduction

Obesity or being overweight are some of the greatest concerns in public health in the world today [1]. It is well known that inhibiting both of the enzymes α-glucosidase and lipase in the digestive system helps to prevent human obesity. Therefore, the prevention of these issues using functional food materials is promising, and Zingiberaceae species have recently become a focus of substantial attention globally in various related research fields [2]. In particular, the *Curcuma* genus, consisting of about 100 species belonging to the Zingiberaceae, is of interest and is widely distributed in tropical regions from Asia to Africa and Australia. Some of these species have been used in folk medicines and as food pigments and occasionally cultivated as ornamental plants. One of the most famous species in this genus is turmeric (*C. longa* L.), which produces and stores huge amounts of curcuminoids in its rhizome. These curcuminoids show various biological activities, and this species is widely cultivated as a health food material [3]. In this study, the enzymatic inhibitions of crude extracts and their constituents from Zingiberaceae against both rat intestinal α-glucosidase and porcine pancreatic lipase were evaluated. Additionally, structure–activity relationships using their derivatives were also investigated. Mango ginger (*Curcuma amada* Roxb.) is a perennial plant with similarly shaped rhizomes to ginger root, but with a mango flavor. The name “mango ginger” is a source of some confusion because this name is used for two species, *C. amada* Roxb. and *C. mangga* Valeton and Zijp, and although they have similar properties and origins, they are distinctly different [4]. The rhizomes are prepared in pickles and drinks in India because mango ginger extract shows various antioxidant, antimicroorganism, and cytotoxic biological activities. The constituents of mango ginger (hereon this indicates *C. amada* Roxb.) rhizome consisted of several labdane diterpenes [5] and monoterpene volatiles, such as myrcene and pinene [6]. Additionally, a biologically active sesquiterpenedimer, difurocumenonol, was isolated and investigated in relation to its accumulation pattern during plant development in mango ginger [7,8]. Other chemical information and biological functions of this plant were previously described in detail [9].

## 2. Results and Discussion

### 2.1. Screening Tests of Zingiberaceae Extracts for α-Glucosidase and Pancreatic Lipase Inhibition

The strongest α-glucosidase inhibitory active extract prepared from selected Zingiberaceae was the ethyl acetate extract of turmeric set as 100% inhibition at 1 mg/mL. The results of screening tests were similar to a previous evaluation of several ethanol extracts prepared from Zingiberaceae and acarbose against α-glucosidase [10]. Although turmeric is well known for its α-glucosidase inhibition activity, we selected other Zingiberaceae species [11]. The efficacy of inhibitors can vary substantially, depending on the origin of α-glucosidase, between yeast and rat intestine. Generally, α-glucosidase from rat intestine is less sensitive to inhibitors, so its IC_50_ values and inhibition rates with inhibitors tend to be lower [12]. In our screening test, because remarkable α-glucosidase inhibition activity was present in the ethyl acetate extract of mango ginger, we selected this species for identification of active ingredients (Figure 1).

Meanwhile, for pancreatic lipase inhibitory activity evaluation, most plant extracts included various fluorescent compounds, so 4-methylumbelliferone (4-MU), the lipase hydrolyzed product, was separated between these fluorescent compounds and 4-MU using high-performance liquid chromatography (HPLC). The strongest inhibition activity occurred in the hexane and ethyl acetate extracts of mango ginger. An extract of turmeric also showed strong pancreatic lipase inhibitory activity (Figure 2). It has been reported that the major constituent of mango ginger, (*E*)-labda-8(17),12-diene-15,16-dial (**1**), was effectively extracted using ethyl acetate as a solvent [13].

### 2.2. Preparations of Test Compounds

The spectra data of compounds **1** and **2** were in good agreement with previous published data isolated from another Zingiberaceae, *Alpinia speciosa* [14]. Compounds **1** and **2** were identified as (*E*)-labda-8(17),12-diene-15,16-dial and (*E*)-15,16-dinorlabda-8(17),11-diene-3-one, respectively. In addition, the spectrum data of compound **3** allowed it to be identified as albicanal (Figure 3) [15]. Seven other derivatives **4**–**10** were obtained using various organic syntheses from parent compound **1** (Scheme 1 and Scheme 2, Table 1).

### 2.3. Evaluation of Tested Terpenes for α-Glucosidase and Pancreatic Lipase Inhibition

The strongest rat intestinal α-glucosidase inhibitory activity was for compound **1**. In addition, compound **2** and a reduced derivative of compound **1**, (*E*)-Labda-8(17),12-diene-15,16-diol (**4)**, showed strong α-glucosidase inhibition. However, the drimane sesquiterpene aldehyde, compound **3**, did not show any such activity (Table 1). There is a report that polygodial, a congener of compound **3**, did not show α-glucosidase inhibition (Figure 3) [16], but another report showed that both compound **1** and polygodial inhibited human 5-lipoxygenase [17]. Additionally, similar labdane diterpenolides from another Zingiberaceae, *Hedychium spicatum* Ham. Ex Smith, also inhibited rat intestinal α-glucosidase but drimene did not [18]. Similarly, labdanes and norlabdanes from *Leonurus japonicus* Houtt. (Lamiaceae) showed inhibition of yeast α-glucosidase [19]. These findings suggested that adequate distance between the drimane skeleton and aldehyde group was important for α-glucosidase inhibition. Other tested derivatives, including acetates **5**–**7** and oxidative derivatives **8**–**10** from compound **1**, showed almost no α-glucosidase inhibitory activity. Nevertheless, more detailed research is needed to determine the mode of action for these terpenes.

Of note, the antimycobacterial activity of compound **1**, compound **4** and 15,16-diacetoxyl-(*E*)-labda-8(17),12-diene **(5)** against *Mycobacterium tuberculosis* showed a similar tendency of α-glucosidase inhibition to a previous study [20]. Other previously demonstrated biological activities of compound **1** include antiplasmodial activity [21], antibacterial activities against both Gram-positive and -negative bacteria, and antifungal activity [22] against *Aspergillus fumigatus* and *Fusarium oxysporum*—this antifungal activity disappeared with the reduction of dial moieties to diol derivatives [23].

The strongest inhibitory activity against porcine pancreas lipase was also found for compound **1**. Recently, this compound and its triazole-appended derivatives on the other aldehyde group of compound **1** showed good pancreatic lipase inhibition without cytotoxicity evaluated in Hep G2 cells [24]. In this study, all analogs **2**–**10** showed moderate inhibitory activity except for the sesquiterpene, compound **3**. Compound **3** decreased pancreatic lipase activity drastically compared with compound **1** (Table 1). This result implied a correspondence between α-glucosidase inhibition and pancreatic lipase inhibition in terms of the distance between the drimane skeleton and the aldehyde group. The critical factor of pancreatic lipase inhibition was reported, based on the structure–activity relationship and a modeling study using a docking model between a ligand and lipase enzyme, in which the lipase inhibitor needed a carbonyl group to make a hydrogen bond, as in the β-lactone of tetrahydrolipstatin (Orlistat) (or a hydrophobic moiety) with a hydrophobic interaction with the lid domain [25]. Similarly, the docking model showed that both 4-MU oleate and biflavone from ginkgo bound to the same catalytic site of the lipase. This model also showed how carbonyl and hydroxy groups of the primary flavone acted to make a hydrogen bond with an amino acid in the catalytic site of lipase, and then the secondary flavone of biflavone played a role as a hydrophobic moiety by maintaining an adequate distance from the catalytic site [26]. These spatial properties are important in order to act as a lipase inhibitor.

There has been a report that compound **1** and zerumin A (**5**) showed strong cytotoxicity against cancer cell lines such as MCF-7, KB and A549. However, (*E*)-15,16-dinorlabda-8(17),11-diene-3-one (**2**) did not show cytotoxicity [27]. This tendency was completely different to α-glucosidase and pancreatic lipase inhibition. Additionally, it was reported the compound **1** showed greater inhibition of COX2 than COX1 [28]. This information implied that mango ginger could be used as an anti-inflammatory agent without side effects.

## 3. Materials and Methods

### 3.1. General Experimental Procedures

The 1D- and 2D-NMR spectra were obtained using a Bruker Avance 400 instrument (Bruker Co. Ltd., Bremen, Germany) with solvent signal as an internal reference. The EI-MS and HR-ESI-MS spectra were measured using JEOL JMS-K9 (JEOL Co. Ltd., Tokyo, Japan) and Waters Q-Tof Premier (Waters Corporation, Manchester, UK), respectively. The HPLC analyses were performed on a Shimadzu VP (Shimadzu Co. Ltd., Kyoto, Japan) equipped with a RF-10AXL fluorescence detector using a l-column ODS (5 µm, ø 4.6 × 250 mm; CERI, Tokyo, Japan). Flash column chromatography was performed on Isolera one (Biotage, Uppsala, Sweden).

### 3.2. Plant Materials

Four *Curcuma* species, *C. zedoaria* (Christm.) Roscoe, *C. longa* L., *C. aromatica* Salisb. and *C. amada* and a Zingiber species, *Z. officinale* Roscoe, were selected for α-glucosidase and pancreatic lipase inhibitory screening tests (Table 2). Mango ginger (*C. amada*) was cultivated in Kerala and Tamil Nadu states in southern India, and harvested during January–March 2017. The harvested rhizomes were dried outdoors. The other plant materials were purchased from Arjuna Natural Ltd. (Cochin, India).

### 3.3. Chemicals

For the bioassays, *p*-nitrophenyl-α-glucose (*p*NPG) and acarbose (Glucobay) were purchased from Wako Pure Chemical Co. Ltd. Tetrahydrolipstatin (Orlistat) was purchased from Sigma-Aldrich Co. LLC.

### 3.4. Preparation of Zingiberaceae Extracts

The dried rhizomes were shredded into small pieces and respectively extracted using organic solvents, hexane, ethyl acetate and methanol to obtain crude extracts. The plant material was extracted with hexane and then separated between extract and plant residue by filtration. Subsequently, the plant residue was respectively extracted with ethyl acetate and methanol to obtain crude extracts. These procedures produced hexane, ethyl acetate, and methanol extracts for each plant material with various yields (Table 2).

### 3.5. Isolation of Constituents from Mango Ginger

The dried rhizomes (500 g) of mango ginger were shredded and extracted twice with hexane (1.5 L, 2 days each) to obtain a crude hexane extract. The crude hexane extract was concentrated under reduced pressure using a rotary evaporator to obtain brown oil (5.5 g yield 1.1%). The crude hexane extract (300 mg) was subjected to silica gel flash chromatography with a hexane/ethyl acetate solvent system as the eluent to afford subfractions 1 and 2. Subfraction 2 (210 mg) was subjected to silica gel flash chromatography with hexane/ethyl acetate (gradient 20–85% *v*/*v* ethyl acetate in hexane) again to obtain compound **1** (28.6 mg). Subfraction 1 (42.6 mg) was subjected to silica gel flash chromatography with hexane/ethyl acetate (gradient 5–35% *v*/*v* ethyl acetate in hexane) again to obtain compound **2** (2.9 mg) and compound **3** (6.4 mg) (Figure 3).

(*E*)-Labda-8(17),12-diene-15,16-dial (**1**): Pale brown oil ^13^C-NMR(CDCl_3_) δ ppm (100 MHz): 197.3, 193.6, 159.9, 148.0, 134.9, 107.9, 56.5, 55.4, 42.0, 39.6, 39.4, 39.2, 37.9, 33.6, 33.6, 24.7, 24.1, 21.7, 19.3, 14.4. EIMS (70 eV) *m*/*z* (rel. int. %), 302 (M^+^, 5.4), 284 (41.4), 269 (39.9), 241 (3.8), 187 (12.9), 137 (55.1), 95 (59.5), 81 (96.9), 69 (100).

(*E*)-15,16-Dinorlabda-8(17),11-diene-3-one (**2**): Pale brown oil ^13^C-NMR(CDCl_3_) δ ppm (100 MHz): 198.1, 148.6, 146.7, 133.6, 108.6, 60.8, 54.5, 42.1, 40.9, 39.3, 36.6, 33.6, 33.6, 27.2, 23.2, 21.9, 19.0, 15.1. EIMS (70 eV) *m*/*z* (rel. int. %), 260 (M^+^, 4.5), 245 (3.5), 217 (7.9), 189 (4.9), 149 (18.5), 137 (23.3), 121 (42.1), 109 (30.2), 95 (28.7), 81(100), 69 (50.6), 55 (30.7).

Albicanal **(3)**: Pale brown oil ^13^C-NMR(CDCl_3_) δ ppm (100 MHz): 203.5, 148.6, 108.0, 55.3, 51.0, 42.0, 39.9, 39.4, 38.9, 37.5, 33.5, 23.9, 21.7, 19.2, 14.6. EIMS (70 eV) *m*/*z* (rel. int. %), 220 (M^+^, 1.2), 219 (8.1), 216 (26.7), 201 (63.0), 190 (25.0), 173 (9.5), 145 (22.6), 137 (73.9), 95 (54.7), 81 (88.8), 69 (100), 55 (50.5).

### 3.6. Derivatization of (E)-Labda-8(17),12-diene-15,16-dial

Compound **1** was the starting material for derivatizations to elucidate the structure–activity relationship based on α-glucosidase and pancreatic lipase inhibition. The reduction toward two aldehyde groups of compound **1** was carried out using NaBH_4_ to obtain the corresponding diol **4**, followed by acetylation of diol **4** using anhydrous acetic acid in pyridine to obtain the corresponding mono- and di-acetylated derivatives **5**–**7** (Scheme 1) [20].

Oxidation toward two aldehyde groups of compound **1** was carried out by Pinnick oxidation to obtain the corresponding monocarboxylic acid, zerumin A **(8)** and dicarboxylic acid **9** [29,30] followed by dicarboxylic acid **9** being methylated by treatment with BF_3_MeOH complex to obtain the corresponding dimethylester **10** (Scheme 2).

(*E*)-Labda-8(17),12-diene-15,16-diol (**4**): colorless oil ^13^C-NMR(CDCl_3_) δ ppm (100 MHz): 148.5, 135.5, 132.3, 107.4, 68.6, 61.5, 57.1, 55.4, 42.1, 39.6, 39.2, 38.1, 33.6, 33.5, 32.8, 24.2, 22.3, 21.7, 19.3, 14.4.

15,16-Diacetoxyl-(*E*)-labda-8(17),12-diene (**5**): colorless oil ^13^C-NMR(CDCl_3_) δ ppm (100 MHz): 171.0, 170.9, 148.4, 34.8, 129.3, 107.5, 77.2, 69.0, 62.6, 56.9, 55.4, 42.1, 39.5, 39.2, 38.0, 33.6, 33.6, 28.1, 24.2, 22.4, 21.7, 21.0, 19.4, 14.4.

15-Acetoxy-(*E*)-labda-8(17),12-diene-16-ol (**6**): colorless oil ^13^C-NMR(CDCl_3_) δ ppm (100 MHz): 171.1, 148.5, 134.1, 131.3, 107.4, 67.8, 63.0, 57.1, 55.4, 42.1, 39.5, 39.2, 38.1, 33.6, 33.6, 28.0, 24.2, 22.2, 21.7, 21.1, 19.4, 14.4.

16-Acetoxy-(*E*)-labda-8(17),12-diene-15-ol (**7**): colorless oil ^13^C-NMR(CDCl_3_) δ ppm (100 MHz): 170.9, 148.4, 134.8, 129.8, 107.4, 69.2, 61.0, 57.0, 55.4, 42.1, 39.6, 39.1, 38.0, 33.6, 33.5, 32.2, 24.2, 22.4, 21.7, 21.1, 19.3, 14.4.

Zerumin A (**8**): colorless oil ^13^C-NMR(CDCl_3_) δ ppm (100 MHz): 193.6, 175.1, 159.3, 148.0, 135.7, 107.9, 56.4, 55.4, 42.0, 39.6, 39.2, 37.8, 33.6, 29.6, 24.6, 24.1, 21.7, 19.3, 14.4, 14.2.

(*E*)-Labda-8(17),12-diene-15,16-dicarboxylic acid (**9**): colorless oil, ^13^C-NMR(CD_3_OD) δ ppm (100 MHz): 176.9, 173.7, 149.6, 145.6, 129.0, 108.3, 58.0, 56.6, 43.2, 40.5, 40.2, 39.0, 35.7, 34.4, 34.1, 25.3, 24.8, 22.2, 20.3, 14.9.

(*E*)-Labda-8(17),12-diene-15,16-dimethylester (**10**): colorless oil ^13^C-NMR(CDCl_3_) δ ppm (100 MHz): 171.2, 147.5, 137.6, 128.5, 123.8, 52.0, 51.9, 51.8, 41.5, 38.8, 36.9, 33.8, 33.3, 33.2, 32.3, 27.5, 21.6, 20.0, 19.5, 18.9.

### 3.7. Evaluation of α-Glucosidase Inhibition

The α-glucosidase activity was calculated by measuring the amount of *p*-nitrophenol hydrolyzed from *p*NPG by α-glucosidase. Tested enzyme was prepared using α-glucosidase included in commercial rat intestinal acetone powder (Sigma-Aldrich). Rat intestinal acetone powder (0.1 g) was homogenized with 0.1 M phosphate buffer (3 mL) at pH 6.9 by sonication in an ice bath. Sonication was performed 20 times each for durations of 10 s. After centrifugation at 7000× *g* for 15 min at 4 °C, the supernatant was separated from the pellet. The supernatant was diluted two-fold with 0.1 M phosphate buffer at pH 6.9. The test sample was dissolved in DMSO and then diluted in the same volume of 0.1 M phosphate buffer. The test sample solution (10 µL) was added to a crude enzyme suspension (40 µL). The solution was preincubated at 37 °C for 10 min, then 2.5 mM of *p*NPG dissolved in 0.1 M phosphate buffer (50 µL) was added to the reaction mixture as the substrate. The enzymatic reaction was carried out at 37 °C for 30 min and the reaction was terminated by the addition of 0.1 M Na_2_CO_3_ aq. (900 µL) [31,32,33]. The absorbance at 405 nm was determined. The inhibition percentage was calculated using Equation (1):Inhibition% = [Abs (test sample) − ABS (sample blank)/Abs (control)] × 100(1)

The acarbose was used as a positive control (Figure 4).

### 3.8. Evaluation of Pancreatic Lipase Inhibition

For pancreatic lipase inhibitory enzymatic assays, a slightly modified method from that described in a previous paper was used [34]. The test sample was dissolved in DMSO and then diluted nine-fold with 13 mM Tris–HCl buffer pH 8.0. The test sample solution (50 µL) was added to 50 unit/mL of porcine pancreas lipase suspension (50 µL). The solution was preincubated at 37 °C for 10 min and then 0.1 mM 4-methylumbelliferyl oleate dissolved in 13 mM Tris–HCl buffer (100 µL) was added to the reaction mixture as a substrate. The enzymatic reaction was carried out at 37 °C for 120 min and the reaction was terminated by the addition of citrate buffer (200 µL). After that, the intensity of fluorescent enzymatic product 4-MU was measured by HPLC with fluorescence detection. The gradient program was 0 min (B 21%), 4 min (B 21%), 7.5 min (B 50%), 10 min (B 50%), 10.01 min (B 100%), 15 min (B 100%), 20 min (B 21%), and 25 min (stop) at a flow rate of 1 mL/min (A, water; and B, acetonitrile). The inhibition percentage was calculated using Equation (2):Inhibition% = [Area (test sample) − Area (sample blank)/Area (control)] × 100(2)

Tetrahydrolipstatin was used as a positive control (Figure 5 and Figure 6).

### 3.9. Statistical Analysis

All data were expressed as mean ± standard deviation (SD) from at least three experiments, and IC_50_ was calculated by probit analysis using R software (ver. R. 2.12.1, R Foundation for Statistical Computing).

## 4. Conclusions

Mango ginger contained large amounts of labdane diterpenes compared with other Zingiberaceae. Additionally, (*E*)-labda-8(17),12-diene-15,16-dial (**1**) the major constituent of mango ginger showed a high enzymatic inhibition slightly less than that of the commercial product, acarbose. Therefore, this plant material might be used as a functional food for the prevention of obesity in the future.

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
