# Peer review of "α-Glucosidase and Pancreatic Lipase Inhibitory Activities of Diterpenes from Indian Mango Ginger (Curcuma amada Roxb.) and Its Derivatives"

_molecules, 2019, doi:10.3390/molecules24224071_

Round 1
Reviewer 1 Report
This paper is focused in an interesting study field, the activities of diterpenes from Indian mango ginger (Curcuma amada Roxb.) and its derivatives; however, this paper has some limitations, so, it cannot be accepted for publication in its present form.
Major comments
Abstract. It is very confusing, the authors must include the background, the method, the principal results and the conclusions. Please do not include the compounds you studied in parentheses, this creates confusion and seems to be references. Introduction. It is necessary to include references of some assertions in the first paragraphs. Also, the authors must to highlight the research "why" and "for what", in addition to pointing out the aim of the study. Results and discussion. Authors should order their main results according to the objective of the research. Please do not include the compounds studied with numbers in parentheses (1, 2, 4-10), this creates confusion. In addition, the authors should point out the limitations of the investigation, for example the sample size and the lack of repetition of the experiments. Material and methods. Please, it is necessary that the authors include the "n", model and characteristics of the study material. Conclusions. Authors should not quote tables in the conclusions.
Author Response
Thank you for your valuable comments and they were very helpful to improve my paper this time.
Reviewer 1
This paper is focused in an interesting study field, the activities of diterpenes from Indian mango ginger (Curcuma amada Roxb.) and its derivatives; however, this paper has some limitations, so, it cannot be accepted for publication in its present form.
Major comments
Abstract. It is very confusing, the authors must include the background, the method, the principal results and the conclusions. Please do not include the compounds you studied in parentheses, this creates confusion and seems to be references.
I added the background as line 11-15 to this abstract
I changed numbering of chemicals and cited appropriate references in abstract.
Introduction. It is necessary to include references of some assertions in the first paragraphs. Also, the authors must to highlight the research "why" and "for what", in addition to pointing out the aim of the study.
I changed the presentation and insert a sentence including the objective of this study.
Results and discussion. Authors should order their main results according to the objective of the research. Please do not include the compounds studied with numbers in parentheses (1, 2, 4-10), this creates confusion. In addition, the authors should point out the limitations of the investigation, for example the sample size and the lack of repetition of the experiments.
I changed numbering of chemicals.
The sample size noted in figure legends and Method section.
Material and methods. Please, it is necessary that the authors include the "n", model and characteristics of the study material.
The sample size noted in figure legends and Method section.
The plant materials were showed in Material and Method, all plant materials commercial products.
Conclusions. Authors should not quote tables in the conclusions.
I deleted the quote from this section.
Reviewer 2 Report
In the abstract I suggest add at first some theoretical background which will introduce to aim of this paper.
The citation format should be corrected according to the Author guidelines
Are Authors able to provide some details of cultivation and herbarium number of used plant material?
More details about extract obtaining procedure should be added
Results should be presented as a mean with SD, not SEM
What about the mechanism of action tested compounds, have Authors any comment to add to this paper about that?
Author Response
Thank you for your valuable comments and they were very helpful to improve my paper this time.
Reviewer 2
In the abstract I suggest add at first some theoretical background which will introduce to aim of this paper.
I added the background as line 11-15 to this abstract
The citation format should be corrected according to the Author guidelines
I improved the reference style along with this journal format, I was checked previous publication of Molecule.
Are Authors able to provide some details of cultivation and herbarium number of used plant material?
The plant materials were showed in Material and Method, all plant materials commercial products.
So these botanical vouchers did not exist and I could not deposit any herbarium. Though all of these plants are commercial product, it did not need this time, I think.
More details about extract obtaining procedure should be added
It is enough to explain that showed in section 4-4 line 195-207, I think.
Results should be presented as a mean with SD, not SEM
I changed these results to SD from SEM.
What about the mechanism of action tested compounds, have Authors any comment to add to this paper about that?
I added the sentence in line 126-127.
Round 2
Reviewer 1 Report
The authors corrected the manuscript considering all comments.